# Landscape of Well-Coordinated Fracture Healing in a Mouse Model Using Molecular and Cellular Analysis

**DOI:** 10.3390/ijms24043569

**Published:** 2023-02-10

**Authors:** Deeksha Malhan, Katharina Schmidt-Bleek, Georg N. Duda, Thaqif El Khassawna

**Affiliations:** 1Experimental Trauma Surgery, Faculty of Medicine, Justus-Liebig University of Giessen, Aulweg 128, 35392 Giessen, Germany; 2Institute for Systems Medicine, Faculty of Human Medicine, MSH Medical School Hamburg, 20457 Hamburg, Germany; 3Berlin Institute of Health, Julius Wolff Institute at Charité—Universitätmedizin Berlin, Augustenburger Platz 1, 13353 Berlin, Germany; 4Berlin Institute of Health, BIH Center for Regenerative Therapies at Charité—Universitätmedizin, Augustenburger Platz I, 13353 Berlin, Germany

**Keywords:** fracture healing, microarray, extracellular matrix, mitochondrion, ribosome, wild-type, bioinformatics analysis, histology

## Abstract

The success of fracture healing relies on overlapping but coordinated cellular and molecular events. Characterizing an outline of differential gene regulation throughout successful healing is essential for identifying crucial phase-specific markers and may serve as the basis for engineering these in challenging healing situations. This study analyzed the healing progression of a standard closed femoral fracture model in C57BL/6N (age = 8 weeks) wild-type male mice. The fracture callus was assessed across various days post fracture (D = days 0, 3, 7, 10, 14, 21, and 28) by microarray, with D0 serving as a control. Histological analyses were carried out on samples from D7 until D28 to support the molecular findings. Microarray analysis revealed a differential regulation of immune response, angiogenesis, ossification, extracellular matrix regulation, mitochondrial and ribosomal genes during healing. In-depth analysis showed differential regulation of mitochondrial and ribosomal genes during the initial phase of healing. Furthermore, the differential gene expression showed an essential role of Serpin Family F Member 1 over the well-known Vascular Endothelial Growth Factor in angiogenesis, especially during the inflammatory phase. The significant upregulation of matrix metalloproteinase 13 and bone sialoprotein from D3 until D21 asserts their importance in bone mineralization. The study also shows type I collagen around osteocytes located in the ossified region at the periosteal surface during the first week of healing. Histological analysis of matrix extracellular phosphoglycoprotein and extracellular signal-regulated kinase stressed their roles in bone homeostasis and the physiological bone-healing process. This study reveals previously unknown and novel candidates, that could serve as a target for specific time points in healing and to remedy cases of impaired healing.

## 1. Introduction

Fracture healing is characterized by a well-orchestrated series of biological events involving cellular recruitment, growth, and differentiation [1]. The interdependent healing events of this healing cascade require a finely balanced interplay between different cell types and regulatory factors. Dysregulation at the cellular or molecular level during fracture healing results in delayed healing or eventually nonunion [2]. Recently established model systems of bone healing allowed researchers to identify and unravel the complexity of molecular signaling involved in fracture healing. 

The temporal and spatial dynamics of fracture healing have only occasionally been analyzed in model systems such as wild-type mice using high throughput data analysis [3,4,5]. Nakazawa et al., (2004) identified upregulated genes in the early stage of fracture healing using microarrays at day 3 only; the upregulation of the candidate gene (periostin) in the following days was confirmed using qPCR. This study revealed the importance of periostin in mesenchymal cell proliferation during the early phase of healing [3]. Khan et al., (2008) used an open fracture model and reported the appearance of leptin as a key element in healing, with a lack of leptin leading to an early arrest in the endochondral ossification phase [4]. The main goal of Bias M et al., (2009) was the use of the microarray technique to explore whether the differentially expressed genes during fracture healing are a reflection of embryonic stem cells [5]. Several other studies and reviews have addressed sex differences in mice [6] or compared healing in the wild-type to that of transgenic models such as low-density lipoprotein receptor-related protein 5 (*Lrp5*) knockout [7]. Despite the invaluable information that such studies provide, investigating longitudinal changes in gene expression and cellular aspects to provide an outline for bone healing remains crucial. Additionally, to our knowledge, no study has combined molecular insight via high-throughput data and histology to address the overlapping healing cascades.

In this study, we employed a detailed molecular analysis of fracture healing in a wild-type mouse model to serve as a blueprint useful for diagnosing molecular dysregulation throughout the healing process. Differential gene expression findings revealed six core biological processes with interlinking gene sets that temporally vary in the expression pattern. The processes encompassed immune response, angiogenesis, ossification, extracellular matrix (ECM) formation, mitochondrial, and ribosomal gene regulation. The mitochondrial marker (GPX1) signal was localized in the vicinity of osteocytes and bone marrow fat cells, suggesting a role for osteocytes and adipocytes in regulating energy metabolism during healing. Further, the study highlights the regulatory role of ECM in fracture healing through changes in osteocyte morphology, collagen fibril orientation, MEPE, and ERK activity. The presented findings demonstrate the relevance of a timed intervention to enhance healing in order for agents to support specific healing phases physiologically and ultimately avoid nonunion.

## 2. Results

### 2.1. Bone Matrix Mineralization Progression across the Time Points of Healing

Callus mineralization and bone bridging signify successful bone healing at the cellular level. Trichrome Masson Goldner stain was used to obtain temporal and spatial distributions of mineralized and nonmineralized bone matrices during the overlapping stages of bone healing (Figure 1). 

This wild-type model showed a typical pattern of bone healing that consists of overlapping phases. The injury initiated an inflammatory reaction, thereby forming a hematoma. The inflammatory reaction was then followed by the formation of cartilaginous calluses as seen in D7 (Figure 1a). The presence of proliferative chondrocytes continuously decreases in favor of hypertrophic chondrocytes, leading to mineralization of the matrix and the formation of hard calluses through D10 to D14 (Figure 1b,c). This is where woven bone forms distally and proximally of the fracture gap and directly on the periosteal surface. Later, the rebuilding of the mechanically stable form of bone occurred from D21 until D28 (Figure 1d,e).

Analysis of vascularization around callus was carried out using Alpha smooth muscle actin (ASMA) IHC (Figure 2a). Histomorphometry showed the largest ASMA positive area at D7, and the lowest at D21. ASMA positive area varied at D10, D14, and D28 with no statistically significant differences. Matrix extracellular phosphoglycoprotein (MEPE) and extracellular signal-regulated kinase (ERK) are important for ECM regulation. Descriptively, MEPE positive signal located in the fracture gap and periosteal region and around bone marrow fat cells at D7. At D10, MEPE signal was seen in proliferative chondrocytes and within bone marrow and later at D14 was seen within bone marrow and bone matrix (Figure 2b). ERK positive signal was seen around the callus border, in chondrocytes and within monocytes at D7; near proliferative chondrocytes and within bone marrow at D10 and in small patches in hypertrophy chondrocytes and bone marrow at D14. Later at D21 and D28, ERK signal was located within bone matrix and around fat cells present within bone marrow at D21 (Figure 2c). Glutathione Peroxidase 1 (*GPX1*) is a mitochondrial gene which acts against oxidative stress to maintain bone homeostasis. GPX1 positive signal was seen in small patches within bone marrow and fracture callus at D7. At D14, the signal was located in proliferative chondrocytes, bone marrow, and bone lining cells. An increasing signal from D14 to D28 within bone marrow and bone matrix around fat cells was seen (Figure 2d). Descriptively in toluidine blue counter-stained sections, Ubiquitin B (UBB) signal was seen in chondrocytes, callus border, and bone marrow at D7 (Figure 2e). Further, UBB as a conserved proteasome complex which regulates cell cycle, was seen at D10 in proliferative chondrocytes and endosteum region. Interestingly, at D14 and D21 UBB positive signals was seen within bone marrow and mineralized callus portion. At D28, UBB signal was located within bone marrow and in newly formed bone (Figure 2e).

### 2.2. Differential Expression of Immune Response-Related Genes during Fracture Healing

The immediate pro-inflammatory response to a fracture is central to initiating the healing cascade, activating multiple inflammatory factors. The closely followed anti-inflammatory cascade is a prerequisite for the formation of an organized soft callus. This study showed the timely formation of soft callus at D7, thereby indicating a successful switch, during the inflammatory reaction, from a pro-inflammatory to an anti-inflammatory signaling at an early time point, possibly D3. Tight regulation of the immune response is essential for successful fracture healing. Therefore, differentially expressed genes (DEGs) involved in the immune response were analyzed. Functional annotations resulted in 15 immune response genes with a cutoff of fold change ≥|2| and *p* ≤ 0.01, which were differentially expressed in at least one time point. Both pro- and anti-inflammatory genes were differentially expressed (Table 1). The study, however, did not include a time point within the first 24 h, which affected the detailed exploration of proinflammatory markers.

Both innate and adaptive immune responses are crucial for the success of bone healing. Overall, the differentially expressed genes involved in immune response showed enrichment in biological processes like antibacterial humoral response and negative regulation of viral processes (Appendix A). Neutrophils and monocytes provide the first line of defense immediately after the injury. The S100 calcium binding protein A8 (*S100a8*) gene is secreted by the cytoplasm of neutrophils and is regulated at D3 and D7. The downregulation of *S100a8* after the early inflammatory phase correlates with the decline in neutrophil activity in the later phases. The upregulation of other neutrophil and macrophage secreted markers like secretory leukocyte peptidase inhibitor (*Slpi*) (upregulated at D10), cathepsin G (*Ctsg*) (upregulated at D10 and D14), lipocalin 2 (*Lcn2*) (upregulated at D10 and D14), peptidoglycan recognition protein 1 (*Pglyrp1*) (upregulated at D10 and D14), and interferon induced transmembrane protein 2 *(Ifitm2*) (upregulated at D10) which conducts clearance of internalized pathogens and apoptotic cells after the inflammatory phase was seen. Moreover, macrophage activation directly regulates the Ficolin B (*Fcnb*) gene to conduct the conversion of cartilage to hard callus. *Fcnb* was significantly upregulated at D10. Dendritic cells secrete genes like Radical S-Adenosyl Methionine Domain Containing 2 (*Rsad2*) to prevent viral infection (downregulated at D21 and D28). The Fc fragment of IgE Receptor 1g (*Fcer1g*) gene (upregulated at D10 and D14) also prevents infection and modulates the adaptive immune response. In contrast, certain innate immune markers like Interleukin 11 receptor subunit alpha 1 (*Il11ra1*) prevent cytokine release and are downregulated at D10. The synuclein alpha (*Snca*) gene, which promotes proinflammatory cytokine release, was downregulated at D10, D21, and D28. The pro-inflammatory reaction, which is ended by D3, correlates with the downregulation of *Snca*.

However, the adaptive immune response works through B and T cells not only to provide defense but also to promote healing. The beta 2 microglobulin (*B2m*) gene can form amyloid fibers and can also prevent bacterial infection. *B2m* was downregulated at D3 and D7, whereas it was upregulated at D10. Adenosine monophosphate deaminase 1 (*Ampd1*), which regulates T cells, was downregulated at D7, D10, and D14. A common protein central to cell-mediated cytotoxicity of natural killer cells and T cells is perforin (*Prf*), which was upregulated at D7. B cells express the placenta-specific 8 (*Plac8*) gene, which was upregulated at D10 and D14. The upregulation of *Plac8* at D10 and D14 was correlated with the bone mineralization phase.

### 2.3. Angiogenesis Regulation Overlaps Temporally with the Inflammatory Response

Differential gene expression analysis showed that twelve DEGs related to angiogenesis were expressed at least at one time point. 

Seven genes were significantly downregulated, and five genes were significantly upregulated (Appendix A). Overall, the differentially expressed genes involved in angiogenesis process were enriched in biological processes such as epithelium migration and tissue migration (Appendix A). Angiogenic factors like vascular endothelial growth factor A (*Vegfa*), pleiotrophin (*Ptn*), and antiangiogenic factor like decorin (*Dcn*), were all downregulated. *Vegfa* and *Dcn* were downregulated at D3 and D7 and correlated to the inflammatory phase. The *Ptn* was downregulated at D3, D7, D21, and D28, which correlates with early and late vascularization. Genes with dual functions like Aquaporin 1 (*Aqp1*), which encodes for cell migration; *Gpx1*, which lessens oxidative stress conditions; matrix metallopeptidase 2 (*Mmp2*), which degrades ECM to improve blood vessel formation; and Mitogen-Activated Protein Kinase 14 (*Mapk14*), which is central to MAPK signaling, were downregulated at D3 and D7. Furthermore, genes that stimulate the angiogenesis process like transforming growth factor beta receptor 1 (*Tgfbr1*) and transforming growth factor beta receptor 2 (*Tgfbr2*) were significantly upregulated at D14 and D10, respectively. Stress conditions activate secreted proteins, acids, and cysteine-rich (*Sparc*), which are important for cell proliferation and development. *Sparc* was significantly upregulated at D10.

Intriguingly, antiangiogenic factors like serpin family f member 1 (*Serpinf1*) and ECM anchorage protein-like Matrix metallopeptidase 9 (*Mmp9*) were upregulated. *Serpinf1* was significantly upregulated at D3 and D7, corresponding to the inflammatory phase. The *Mmp9* was significantly upregulated at D10, D14, and D21, corresponding to the integration of blood vessels into the new bone matrix (Figure 3a). 

Angiogenesis and vasculogenesis are parts of neovascularization. On the one hand, angiogenesis refers to the process through which mature differentiated endothelial cells (EC) move from their basement membrane and proliferate to generate sprouts from parental vessels [8,9,10]. On the other hand, bone marrow-derived endothelial progenitor cells, which circulate to sites of neovascularization where they develop in situ into mature ECs, participate in vasculogenesis [8]. Therefore, ASMA IHC staining was used as a marker for EC and indicative of angiogenesis [11,12]. Morphologically identifiable blood vessels were counted, and positive ASMA areas indicating EC were evaluated. We observed changes across time-points in the overall distribution of ASMA positive areas around the fractured callus (Figure 3b).

Histomorphometry showed a higher ASMA-stained area at D7 compared with other time points (Figure 3b,c). The blood vessel count was significantly higher at D7 and significantly lower at D21 (Figure 3d). The higher ASMA activity correlates with the angiogenesis-related gene expression at early time points. At the molecular level, the *Asma* gene was not significantly expressed.

### 2.4. At D10 and D14, a Significantly Higher Number of Ossification Genes were Expressed within the Fracture Callus

Ossification is an important process for transitioning soft callus to hard callus in fracture healing. The coordinated gene expression mediates successful new bone formation, which further reflects bone matrix integrity. Therefore, DEGs involved in ossification were studied. 

In total, thirteen genes related to ossification were differentially expressed at least at one time point; out of which only two genes were downregulated (Appendix A). The differentially expressed genes involved in the ossification process were enriched in biological processes such as biomineralization and cartilage development (Appendix A). Genes that encode chondrocyte hypertrophy like Ubiquitin b (*Ubb*, at D3 and D7) and myocyte enhancer factor 2c (*Mef2c*, at D10 and D14) were downregulated. Collagen family genes like collagen type 1 alpha 1 (*Col1a1*), collagen type 10 alpha 1 (*Col10a1*), and collagen type 2 alpha 1 (*Col2a1*) that are important for bone mineralization were mainly upregulated at D10 and D14. Furthermore, Mmp’s family genes like *Mmp9* and *Mmp13*, which play important roles during bone formation and remodeling besides angiogenesis, were also upregulated. *Mmp9* was upregulated at D10, D14, and D21. *Mmp13* was upregulated from D3 until D21. The tissue inhibitor of metalloproteinase 1 (*Timp1*) was upregulated at D3, D7, and D21. Endochondral ossification-related genes like Cathepsin k (*Ctsk*, which was upregulated at D10, D14, and D21) and specificity protein 7 (*Sp7*, which was upregulated at D3, D7, and D14) were significantly upregulated. Further, matrix gla protein (*Mgp*), which inhibits bone mineralization, was significantly upregulated at D10. Secreted phosphoprotein 1 (*Spp1*), which stimulates bone formation and osteoclast anchorage, was significantly upregulated at D10 and D14. Integrin-binding sialoprotein (*Ibsp*), which binds to hydroxyapatite and provides stiffness and strength to bones, was significantly upregulated from D3 until D21.

The temporal distribution of ossification genes correlated with the required phases of healing. Furthermore, bone strength is governed by the arrangement of collagen fibrils and ECM integrity. Therefore, histological analysis of ECM markers was needed.

### 2.5. Histological Analysis of Collagen Fibers Depicted a Cross-Talk between Ossification and ECM Regulation

Osteocytes and type I collagen are the major components of bone matrix and determinants of its integrity and, thus, its biomechanical competence. In physiological bone, osteocytes contribute to bone homeostasis. However, upon bone matrix damage, osteocyte apoptosis is induced [13], which promotes bone resorption [14,15] and formation [16]. Osteocyte activity is usually described through their morphological differences. Morphologically, osteocytes are categorized as empty lacunae (dead), intermediate (spherical), or spindle (active). The changes in osteocyte morphology during the cascade of healing can help unveil the underlying mechanism. Therefore, morphological changes in osteocytes were examined during fracture healing (Figure 4a–d). All three morphological types of osteocytes were seen around the fracture callus at all-time points (Figure 4a). A quantitative evaluation of the osteocyte count in the fracture callus showed a higher empty lacuna count at D7, which correlated with injury and cell apoptosis (Figure 4b). Although the empty lacunae were distributed within the matrix, they were more abundant at the cortices closer to the gap at D7 and D10. Later in D28, the empty lacunae were located more frequently along the cortical bone and the outer cortex of the callus. The absolute count of empty lacunae was significantly higher at D7 (mean ± SEM = 10.94 ± 1.073; *p* = 0.009), D10 (8.76 ± 0.638; *p* = 0.039), and D28 (7.59 ± 0.476; *p* = 0.032) when compared to D14 (6.15 ± 0.781). D28 showed a significantly lower empty lacunae count than D10 (*p* = 0.029) and D21 (*p* = 0.029), when normalized to the bone area. In the case of spherical (intermediate) osteocytes, the absolute count was significantly higher at D7 (8.60 ± 0.840) compared to D14 (5.68 ± 0.636; *p* = 0.039) and D28 (4.96 ± 0.350; *p* = 0.001). The spherical osteocyte count normalized to the bone area was higher at D7 compared with other time points (Figure 4c). While D28 showed lower spherical osteocytes compared to D7 (*p* = 0.045), D10 (*p* = 0.001), and D21 (*p* = 0.008). The absolute count of spindle (active) osteocytes was significantly higher at D14 (18.64 ± 0.870; *p* = 0.004) and D21 (18.67 ± 0.802; *p* = 0.003), compared to D7 (15.27 ± 0.919). The spindle osteocytes count was highest at D7 compared with other time points (Figure 4d). However, D28 showed significantly fewer spindle-shaped osteocytes than D7 (*p* = 0.045) and D10 (*p* = 0.002).

Type I collagen fibers were seen in the soft callus and around the cortices at D7, corresponding with the early ossification phase. Type I collagen at D10 was observed in the periosteal region, corresponding to intramembranous ossification. Type I collagen distribution changed with the healing progression. A different orientation of fibrils was seen at D14, with progression toward endochondral ossification and hard callus formation. Well-aligned and longer collagen fibers were seen in both the endosteum and periosteum regions at D21 and D28. The collagen fibers were seen in the fractured callus region at all time points (Figure 4e). Intriguingly, type I collagen rings were seen around the osteocytes at all time points (Figure 4f). 

Polarized microscope monochrome images of Sirius red-stained sections were also examined using the CT-FIRE plug-in (Figure 5). The plug-in quantifies the properties collagen fibers (fiber angle, fiber length, fiber width, and fiber straightness). The fiber angle was significantly higher at D28 compared with D7 (*p* = 0.013) (Figure 5c). Fiber length was significantly higher at D28 compared with D7 (*p* = 0.005), D10 (*p* = 0.014), D14 (*p* = 0.007), and D21 (*p* = 0.001) (Figure 5d). Fiber width was significantly higher at D28 compared with D7 (*p* = 0.042) and D21 (*p* = 0.015) (Figure 5e). Fiber straightness showed no significant differences between the time points (Figure 5f). The collagen fibril properties also correlated directly with the differential expression of the *Col1a1* gene, which was upregulated at D10, D14, and D21.

The cellular investigation of the bone matrix showed varied changes in osteocyte morphology and collagen arrangement. A molecular understanding of ECM genes is further needed to unravel the complexity of fracture healing. 

### 2.6. ECM Is a Central Regulator in the Maintenance of Bone Homeostasis

ECM serves as a natural reservoir for growth factors and cytokine activity. The well-coordinated molecular and cellular regulation of the ECM helps maintain bone homeostasis. Therefore, DEGs involved in ECM regulation were investigated. 

Twenty-nine ECM-related genes were differentially expressed at least one time point, out of which only six were downregulated (Appendix A). ECM degradation occurs at the early phase of healing to create a passage for new bone formation. While ECM formation and integrity are established during the intermediate and late phases of healing. Overall, the differentially expressed genes were found to be enriched in ECM organization and collagen fibril organization (Appendix A). Lysyl oxidase (*Lox*), which plays a role in collagen’s posttranslational change, was significantly downregulated at D3 and significantly upregulated at D10 and D14. The upregulation of *Lox* correlates with the intermediate healing event where new bone formation occurs. ECM degradation markers like *Mmp2* and ECM deposition factor like osteoglycin (*Ogn*) were significantly downregulated at D3 and D7. Proteoglycans like *Dcn* and lumican (*Lum*) that are important for ECM structural integrity were also downregulated at D3 and D7. collagen type 12 alpha 1 (*Col12a1*) was significantly downregulated at D7. The *Ptn* gene interacts with mesenchymal stem cells and was significantly downregulated at D3, D7, D21, and D28. ECM markers like *Mmp13* and *Mmp9* are crucial for cell differentiation and invasion, besides their importance in angiogenesis. *Mmp13* was upregulated from D3 until D21. *Mmp9* was significantly upregulated at D10, D14, and D21. Genes like *Ibsp* and *Timp1* that regulate ECM deposition were also differentially expressed during fracture healing. *Ibsp* was upregulated from D3 until D21, while *Timp1* was significantly upregulated during D3, D7, and D21. Genes like *Tgfbr1* and *Tgfbr2*, which are important for TGF-beta signaling besides angiogenesis, were upregulated at D14 and D10, respectively. Sh3 and Px domains containing protein 2b (*Sh3pxd2b*) and epidermal growth factor containing fibulin ECM protein 2 (*Efemp2*), which play a role in cell-matrix adhesion, were significantly upregulated at D3 and D7. *Mgp*, which helps in ECM modeling besides ossification, was significantly upregulated at D10. 

Gene-like tenascin c (*Tnc*), which helps in ECM development, was significantly upregulated at D10 and D14. ECM structural components like aggrecan (*Acan*) and ECM degradation markers like cartilage-associated protein (*Crtap*) were significantly upregulated at D10. Carboxypeptidase Z (*Cpz*) is important for binding to Wnt and was significantly upregulated at D10 and D14. Many collagen family genes, like collagen type 24 alpha 1 (*Col24a1*), collagen type 6 alpha 1 (*Col6a1*), *Col1a1*, *Col1a2*, *Col10a1*, collagen type 9 alpha 2 (*Col9a2*), and *Col2a1*, were significantly upregulated during healing. Gene-like *Sparc*, which is associated with collagen fiber assembly, was significantly upregulated at D10.

The differential expression analysis of ECM proteins showed an overlapping role with ossification. Furthermore, bone homeostasis relies on balanced osteoblast and osteoclast activity. Therefore, signals of osteoblastogenesis markers, such as ERK and osteoblast and osteoclastogenesis inhibitor MEPE, were explored through immunostaining (Figure 6). Overall, MEPE and ERK IHC showed changes in the positive signal depending on the time points of healing (Figure 2b,c). At the molecular level, the *Erk* and *Mepe* genes were not differentially expressed.

The MEPE-positive signal was seen in the fracture gap and in the periosteal region at D7. The MEPE signal was seen in proliferative chondrocytes and within bone marrow at D10. The MEPE signal at D14 was seen within the bone marrow and bone matrix (Figure 6a). The MEPE signal at D21 was seen in osteocytes and within the bone marrow. Further, at D28, the MEPE signal was seen within the bone marrow surrounding fat cells. Quantitatively, a higher MEPE-positive staining area was seen at D7 when compared with D10, D21, and D28 (Figure 6b). However, D10 showed a higher MEPE staining area than D21 and a lower stained area than D7, D14, and D28. The MEPE-staining area was highest at D14 (Figure 6b). At D21, the lowest MEPE-positive area was seen; however, it was higher at D28 compared with D10 and D21 (Figure 6b) and significantly lower at D10 compared with D28 (*p* = 0.038). 

An ERK-positive signal was seen around the callus border, in chondrocytes, and within monocytes at D7. An ERK-positive signal at D10 was seen near proliferative chondrocytes and within the bone marrow. D14 showed small patches of ERK signaling in hypertrophic chondrocytes and bone marrow. Further, an ERK signal was seen within the bone matrix and around bone marrow fat cells at D21 (Figure 6c). ERK-positive signal was seen in the bone matrix, in the bone marrow, and close to the newly formed bone at D28. Histomorphometry showed a higher ERK-stained area at D7 compared with D10 and D14 (Figure 6d). Additionally, the ERK staining area was lower at D10 compared with other time points. D14 showed a higher area than D10 (Figure 6d). The ERK staining area was higher at D21 compared with other time points. D28 showed a higher positive area than D7, D10, and D14 (Figure 6d). Further, the ERK-stained area was significantly lower at D10 compared with D21 (*p* = 0.038) and D28 (*p* = 0.009).

### 2.7. Downregulation of Mitochondrial Genes seen at D3 and D7

Regenerative processes require energy regulation, and mitochondria are the source of energy. Therefore, mitochondrial activity-related genes were investigated. 

Thirty-nine mitochondrial genes were differentially expressed at least at one time point, out of which only one gene was significantly upregulated (Appendix A; Figure in Section 2.8). Most genes were downregulated at the early phase of healing, and the genes were enriched in biological processes like aerobic respiration and mitochondrial transport (Appendix A). The hypoxic conditions, at the time of injury, limit the source of energy required for fracture healing. Genes like *Gpx1*, Peroxidase 2 (*Prdx2*), and Peroxidase 3 (*Prdx3*) that play roles in oxidative stress conditions to overcome hypoxia were downregulated at D3 and D7. 3-Oxoacid CoA transferase 1 (*Oxct1*) plays a central role in ketone body catabolism and was also downregulated at D3 and D7. Genes like Solute Carrier Family 25 member 11 (*Slc25a11*), Solute Carrier Family 25 member 20 (*Slc25a20*), and Branched chain ketoacid dehydrogenase kinase (*Bckdk*) that play a role in mineral transport and exchange were downregulated at D3 and D7. Other genes, like transmembrane protein14C (*Tmem14c*), ATP synthase H+ transporting mitochondrial- f1complex and barn complex (*Atp5f1* and *Atp5b*), and translocase of outer mitochondrial membrane 7 (*Tomm7*), that are important for mitochondrial transport and the electron transport chain, were downregulated at D3 and D7. Genes, like citrate synthase (*Cs*), fumarate hydratase I (*Fh1*), succinyl CoA ligase (*Suclg1*), ubiquinol-cytochrome C reductase core protein 2 (*Uqcrc2*), and *Cyc1*, that are important for signaling pathways and coordination of energy production, were downregulated at D3 and D7. 

Coenzyme Q9 (*Coq9*), which involves lipid biosynthesis for the electron transport chain, and cytochrome P450 family 27 subfamily A member 1 (*Cyp27a1*), which plays a role in maintaining cholesterol homeostasis, were downregulated at D3 and D7. Further, erythropoiesis regulatory genes such as heat-shock protein family A member 9 (*Hspa9*), and metal binding genes such as iron-sulfur cluster assembly 1 (*Isca1*), were downregulated at D3 and D7. The Cold Shock domain containing protein E1 (*Csde1*) acts as an RNA-binding protein during transcription/translation and was also downregulated at D3 and D7. Glucose and fatty acid metabolism is important for bone homeostasis, and they are regulated by genes like pyruvate dehydrogenase kinase 4 (*Pdk4*). *Pdk4* is regulated at D3 and D7. Amino acid metabolism genes like electron transfer flavoprotein alpha subunit (*Etfa*) were downregulated at D3 and D7. Furthermore, glutathione S transferase P1 (*Gstp1*), which links stress kinase and the cell apoptotic pathway, was downregulated at D3 and D7. Another downregulated gene at D3 and D7 was translocase of inner mitochondrial membrane 8 homolog B (*Timm8b*), which acts as a chaperone for protein transport. Genes like Voltage Dependent Anion Channel 1 and 3 (*Vdac1* and *Vdac3*) that perform roles in diffusion and binding of molecules were also downregulated. *Vdac1* was downregulated at D7, whereas *Vdac3* was downregulated at D3, D7, and D21. Further, carnitine O-palmitoyltransferase 1 (*Cpt1b*), which acts as a unit for fatty acid beta-oxidation, was regulated at D10 and D14. Other downregulated genes were cytochrome c oxidase subunits 8b, 7a1, 6a2 (*Cox8b*, *Cox7a1*, and *Cox6a2*) that conduct electron transport activity. *Cox8b* and *Cox7a1* were downregulated at D10 and D14. *Cox6a2* was downregulated only at D10. Another important gene was creatine kinase S-type (*Cktm2*), which serves as an energy transducer. *Ckmt2* was downregulated from D3 until D14. Further, a carbohydrate metabolism gene; succinate dehydrogenase cytochrome b560 subunit (*Sdhc*), was downregulated at D14. The Lyr motif-containing protein 5 (*Lyrm5*) gene acts as an electron transfer flavoprotein regulator and is downregulated at D14. Another important gene was Dual Specificity Phosphatase 26 (*Dusp26*), which acts as an inhibitor of Mapk1 and Mapk3. *Dusp26* was downregulated at D14. Heme biosynthesis gene; solute carrier family 25 member 37 (*Slc25a37*) was significantly downregulated at D21 and D28. 

The only upregulated gene was a molecule exchange factor like Solute carrier family 25 member 5 (*Slc25a5*), which was significantly expressed at D10.

The intriguing downregulation of mitochondrial genes during the early phase of healing urged the examination of mitochondrial activity at the cellular level. As a cytosolic enzyme that is expressed in most cell types, the antioxidant protein (GPX1) was chosen for further IHC analysis (Figure 7). Later, the positive signal localization of GPX1 needed to be examined in different cell types throughout the healing progression. 

IHC of GPX1 was performed with toluidine blue counterstaining for overall evaluation across the healing events (Figure 3d). Descriptively, a positive GPX1 signal was detected in small batches within the bone marrow and in the callus region at D7. Furthermore, at D10, the GPX1-positive signal was clearer in proliferative chondrocytes, bone marrow, and bone lining cells. A positive signal at D14 was seen within the bone marrow and bone matrix. Furthermore, at D21 and D28, a higher GPX1-positive signal was detected within the bone matrix and around bone marrow fat cells. Additionally, a GPX1 signal at D28 was also seen around the woven bone (Figure 7a). Histomorphometry showed a lower GPX1-positive area at D7 compared with other time points (Figure 7e). D28 showed a higher GPX1-stained area than other time points (Figure 7e). However, no statistically significant differences were found. 

To further explore the cell type-specific changes in GPX1 activity during bone healing, silver nitrate was used as a counterstain. Silver nitrate helped in the visualization of the GPX1 signal within osteocytes and around blood vessels using fluorescence (Figure 7c). A higher GPX1-positive signal was seen around blood vessels than osteocytes at D7. However, at D10, a GPX1-positive signal appeared in patches within osteocytes and in their vicinity, besides being localized around blood vessels. Interestingly, at D14, the GPX1-positive signal became more localized within osteocytes than in blood vessels (Figure 7c). However, at D21 the GPX1-positive signal was seen in the osteocytes’ vicinity and around blood vessels. Whereas no GPX1 specific signal was detected within osteocytes at D28.

The molecular and cellular changes observed in the mitochondrial genes encouraged further investigation of ribosomal activity during fracture healing.

### 2.8. Ribosomal Genes Are Significantly Expressed during the Early Inflammatory Phase

The ribosome acts as a source of protein synthesis and translation, which is a crucial part of the regeneration process. Therefore, the sixth regulated biological process, which involves ribosomal activity, was studied. 

At least fifteen ribosomal genes were differentially expressed at one time point, out of which only two were upregulated (Appendix A). Ribosome biogenesis and cytoplasmic translation were among the top ten enriched biological processes in differentially expressed genes (Appendix A). Cell differentiation and proliferation markers like *Rps6*, ribosomal protein L24 (*Rpl24*), ribosomal protein L27 (*Rpl27*), ribosomal protein L31 (*Rpl31*), Rps2, and ribosomal protein L36a-like (*Rpl36al*) were downregulated at D3 and D7. Ubiquitin B (*Ubb*) regulates the cell cycle by the proteasome complex and is a crucial gene for cell survival. *Ubb* was downregulated at D3 and D7. Furthermore, mitochondrial ribosomal proteins L53 and L33 (*Mrpl53* and *Mrpl33*) were downregulated at D3 and D7. The first plays a role in erythrocyte differentiation, and the latter regulates ribosomal constituents. The ribosomal biogenesis process was affected by the downregulation of nucleophosmin 1 (*Npm1*) at D3 and D7. Methionyl aminopeptidase 1 (*Metap1*), which plays a role in protein maturation, was downregulated at D7. Interestingly, E2F transcription factor 2 (*E2f2*), which controls cell cycle progression, was downregulated at D21 and D28. Intriguingly, important RNA splicing genes such as heterogeneous nuclear ribonucleoprotein H1 (*Hnrnph1*) were downregulated at D3, D7, D21, and D28. Interestingly, genes like metallothionein 3 (*Mt3*), which encodes ion homeostasis and protein disulfide isomerase family member 3 (*Pdia3*) which encodes the endoplasmic reticulum, were significantly upregulated. *Mt3* was significantly upregulated at D10, D14, and D21. *Pdia3* was significantly upregulated at D10.

The unexpected downregulation of ribosomal genes in the early time points encouraged further investigation of cellular changes related to ribosomal activity in fracture healing. In addition to autophagy, the ubiquitin-proteasome complex is the second major pathway to eliminate aberrant proteins. Therefore, a conserved proteasome complex (UBB) was localized using IHC (Figure 7). The positive signal of UBB was examined throughout the fractured callus (toluidine blue counterstaining; Figure 3e) and specifically in the osteocyte vicinity (silver nitrate counterstaining and fluorescence detection). 

A UBB-positive signal was seen in chondrocytes, the callus border, and bone marrow at D7 in the toluidine blue counterstained sections (Figure 7b). Furthermore, UBB-positive signal at D10 was seen in the proliferative chondrocytes and endosteum region. Interestingly, both D14 and D21 showed an UBB-positive signal within the bone marrow and mineralized callus region. A UBB-positive signal was seen within the bone marrow and in a newly formed bone at D28. Histomorphometry showed a higher UBB-stained area at D7 compared with other time points (Figure 7f). A UBB-stained area was significantly higher at D7 compared with D21 (*p* = 0.003) and D28 (*p* = 0.006). Also, D10 showed a significantly higher UBB-stained area compared with D21 (*p* = 0.040). D14 showed a significantly higher UBB-stained area compared with D21 (*p* = 0.017).

UBB counterstained with silver nitrate showed more positive signals around blood vessels than osteocytes (Figure 7d). The UBB signal was prominent around the blood vessels at D7. The UBB signal at D10 was lower around blood vessels compared with D7. Furthermore, the UBB signal at D14 around blood vessels increased compared with D10. D21 and D28 showed rather fewer positive signals around blood vessels. No osteocyte-specific positive signals were seen. 

The functional annotation of genes resulted in six representative biological processes identified here: immune response, angiogenesis, ossification, ECM regulation, mitochondrial activity, and ribosomal activity. These processes appear fine-tuned in successful fracture healing, which relies on a well-coordinated dynamic of the underlying gene expression (Figure 8).

## 3. Discussion

Increased preclinical and clinical studies over the last decade has contributed to better understanding of the successful process of bone regeneration. Such insight helped in designing therapeutics targeting key processes to remedy nonunion or delayed healing. However, the complete and healthy bony consolidation of nonunion or pseudoarthrosis patients remains an important clinical concern. Nonetheless, healing improvement cannot be achieved without recognizing the molecular regulation underlying the overlapping healing cascade. Therefore, this study illuminated the underlying molecular mechanisms and cellular crosstalk during overlapping healing events in a wild-type mouse model. Histological analysis showed bony consolidation and bridging of cortices 28 days postfracture. Whole genome analysis using microarrays and the following functional annotation of genes resulted in six representative biological processes. The role of the first four — immune response, angiogenesis, ossification, and ECM regulation — in fracture healing was addressed in several previous studies [4,5,17,18,19,20,21]. However, the role of the other two, mitochondrial and ribosomal activities, in fracture healing has not yet been explored. 

Although the study reflects the overlapping nature of bone healing phases, the main biological processes are regulated concurrently in every phase. Moreover, key genes that are well known through histology and previous studies to regulate one process are not the sole regulators of the process. Therefore, the direction of their regulation alone must not decide the activity of the process in a given healing phase. For example, the importance of angiogenesis during the early inflammatory phase of the bone regeneration process is ample [22,23,24,25]. Furthermore, failure in angiogenesis during the early phase is reported to result in atrophic nonunion [26]. VEGF protein is a historically known angiogenic factor activated under hypoxic conditions and it stimulates bone regeneration [27,28]. The reason for the downregulation of *Vegfa* during the inflammatory phase of this study is unclear. However, the synergistic effect between the immune response and VEGF is well documented [29,30]. Therefore, we speculate that Vegf was expressed at an earlier timepoint and initiated the immune response. However, evidence also suggests that a lower bone formation rate is associated with lower VEGF expression during altered oxidative stress [31]. This is consistent with the downregulation of oxidative stress-resistant genes such as *Gpx1*. Interestingly, lower mitochondrial genes leading to oxidative stress and hypoxia were reported as important for the recruitment and migration of stem cells in an in vitro fracture model [32]. In other words, the data suggest that an early upregulation of Vegf could have regulated the immune response, which was further regulated by the hypoxia marked by downregulation of antioxidant genes such as *Gpx1* and *Ubb*. Nevertheless, angiogenesis requires vascular growth and ECM degradation to occur. The ECM degradation nictitates the degradation of multiple enzymes, most crucially Mmps [33]. *Serpinf1* upregulation at the early phase of healing (i.e., D3 and D7) followed by *Mmp9* upregulation at D10, D14, and D21 is important for blood vessel formation and continues bone mineralization [34,35]. Furthermore, dysregulated expression of Serpinf1 or the loss of Serpinf1 is reported to result in osteogenesis imperfecta (OI) type VI [36]. Additionally, patients with OI show deficiencies in bone mineralization and collagen fibers [37,38]. Previous studies also showed an improvement in bone healing in a murine model through VEGF administration when applied at the later stages of nonunion [39]. Moreover, the slow release of VEGF at the site of bone damage showed the effectiveness of VEGF during endochondral ossification [27]. Also, dual delivery of VEGF and BMP2 showed a synergistic effect on bone formation in a rat critical size defect model [40]. However, Serpinf1 was reported to play a pivotal role in neovascularization, and its loss results in the negative regulation of angiogenesis [41]. This suggests that further investigation of the neovascularization impairment as a cause of impaired healing should address vascular growth and ECM degradation simultaneously, in the contexts of immune response and oxidative stress.

Furthermore, the fact that the early inflammatory phase influences the bone healing process is well known. However, the differential regulation between angiogenesis genes and pro- and anti-inflammatory cytokines is a subject of many current studies. In this study, investigation of immune response genes showed the upregulation of both pro- and anti-inflammatory genes during the reparative phase. One important example is the downregulation of the proinflammatory gene (*B2m*) at D3 and D7 and the upregulation at D10. This expression pattern suggests that inhibiting *B2m* during the early inflammatory phase and activating *B2m* during the reparative phase can be more beneficial in enhancing healing. Previous studies reported increased serum levels of B2m in postmenopausal osteoporosis [42], Paget’s disease [43], and rheumatoid arthritis [44]. Therefore, a specific release of B2m is crucial to prevent the risk of inflammatory and bone diseases. Also, *B2m* is of particular interest because of immune cell involvement in the regulation of bone turnover. Moreover, network analysis revealed the co-expression interaction between *B2m*, *Mmp9*, and *Serpinf1*. Therefore, inhibiting *B2m* in the early phase might also reflect on increased *Serpinf1* expression for successful angiogenesis and bone mineralization through *Mmp9*. Such phase specific activity of pro-/anti-inflammatory genes along with angiogenesis might be advantageous toward the success of the early phase. 

The successful early inflammatory phase affects the following phase, like ossification, which is crucial for bone matrix integrity. Many key factors like *Mmp13*, *Mmp9, Ibsp*, and collagens are important in both ossification and ECM regulation. Nevertheless, this study suggests that knowing the temporal distribution of genes essential for both ossification and ECM regulation is crucial to predicting their therapeutic potential. Our data suggests the use of more than one candidate marker to tackle ossification and ECM regulation, especially biomarkers that have no fluctuating gene expression pattern throughout the healing. For instance, we highlighted *Ibsp* and *Mmp13*, which are involved in ECM regulation and ossification from D3 to D21. Despite the regulation of two different pathways (focal adhesion pathway and IL-17 signaling, respectively), recent reports showed that delayed ossification and undermined growth occur in the absence of either Ibsp [45] and/or Mmp13 [46]. Therefore, more research is needed to determine the benefits of using two functionally distinct factors rather than one pathway-based factor that accounts for the same histopathological appearance as a preventive therapy for anticipated delayed healing. Using pre-treatment screening to determine the genetic cause is not feasible. 

The orchestrated gene regulation during ossification and ECM regulation was reflected in the main components of bone tissue, type I collagen, and osteocytes. In this study, quantitative evaluation of collagen fibrils showed higher fiber length, which pointed to mechanical stability at later time points (Figure 5). This accords with the reported accumulation of smaller fibrils in inferior quality bones, such as in the biopsies of patients with multiple tumors [47] and osteoporotic rat models [48]. Furthermore, discrepancies in collagen fibril properties have been reported under different metabolic diseases and in aged conditions [49,50]. Osteocyte morphology and count mark the inferior quality of bone tissue. Moreover, osteocytes are among the master instigators of bone mineralization. The arrangement of type I collagen fibers around spindle-shaped osteocytes indicated that the deposition of new mineralized bone occurs first along the canaliculi network. This observation aligns with the previous studies that showed how, with the maturation of osteocytes, the vacated volume is replaced by minerals and osteocytes conduct direct matrix synthesis [51]. During the healing process, new bone formation and enhanced biomechanical stability are needed. In this study, the increase in spindle-shaped osteocyte count and decline in empty lacunae count with the healing progression indicated improved mechanical stability. This is in line with previous studies that showed how the discrepancies in the osteocyte count are correlated to bone fragility and lower biomechanical competence in inferior osteoporotic bone [52,53].

Bone matrix integrity is affected by mechanical stimuli and osteoblast proliferation through MAPK signaling, which plays a role in cell interaction and proliferation [54]. MEPE is involved in regulating bone formation and bone mineralization through ERK activation in osteoblasts [55]. Indeed, in this study, immunohistochemical evaluation of MEPE and ERK signals showed a feedback correlation between them across the time points of healing. Previous studies have reported that MEPE inhibits osteoclastogenesis [56], while ERK positively regulates osteoblastogenesis [57]. Therefore, MEPE can be a possible therapeutic target to prevent osteoclastogenesis in diseases like osteoporosis and osteomalacia. Moreover, MEPE and ERK together can be used to prevent osteoclastogenesis and osteoblastogenesis in diseases like Paget disease. Nonetheless, higher MEPE signals at D7, D14, and D28 (Figure 6b) and higher ERK signals at D21 and D28 (Figure 6d) suggest their overlapping roles in the bone remodeling phase. The higher signal of MEPE at D7 suggested the subsequent activation of ERK to conduct bone mineralization at D10 and D14. Nevertheless, higher MEPE and ERK signals at D28 suggest their combined potential in the bone remodeling phase to preserve homeostasis and prevent bone loss. Moreover, the network analysis showed the co-expression interaction between *Mepe*, *Ibsp*, *Mmp13*, and *Mmp9* that work together to maintain ECM.

ECM is central to recruiting immune cells upon injury or in the case of infection [58]. Several cytokines and growth factors bound to the ECM are released by matrix proteases and influence immune cell proliferation and differentiation [59,60,61]. For example, TGF-B regulates innate and adaptive immune cell function and is released from the ECM by MMPs upon injury or infection [62,63]. Interestingly, our results explore the importance and pattern of innate and adaptive immune responses in fracture healing. For example, the downregulation of *B2m* showed an inverse relation with the immune system. This observation accords with previous studies that showed higher serum levels of B2m among patients with rheumatoid arthritis and multiple myeloma [64]. Therefore, *B2m* might serve as a therapeutic marker, indicating a dysregulation of the immune response during bone healing. Intriguingly, immune dysfunction has also been seen among patients with mitochondrial disorders [65,66]. 

Although oxidative injury is described in osteoporotic patients [67], and as a complication of traumatic injuries to the brain and spinal cord [68], it is not the focus of fracture healing studies. Various studies have indicated the therapeutic potential of targeting oxidative stress with Gpx1 in multiple organs and diseases [67,68,69,70].

The balance required between hypoxic conditions to promote stem cells and angiogenesis [31,32] on the one hand and protection against oxidative stress on the other hand is critical for physiological bone healing. 

The data suggest that such a balance could have been achieved within the fracture callus through the selective expression of genes in different cell types. Interestingly, despite the global downregulation of *Gpx1,* the histological analysis of fracture callus in this study indicates higher GPX1 signal intensity in osteocytes across all time points (Figure 7a,c,e). 

In addition to osteocytes, a GPX1-positive area was seen in the bone marrow fat cells. *Gpx1* is co-expressed with PPAR coactivator 1 alpha (*Pgc1a*) gene, which is a brown/beige adipocyte-specific marker and is important for the energy metabolism and mitochondrial biogenesis [71,72]. This suggests a cross-talk between GPX1 and PGC1a during the early phase of healing. However, *Pgc1a* gene differential expression was only visible in our data when a less stringent cut-off (*p* ≤ 0.05) was applied.

Furthermore, the activation of TGF-BR1 and ERK as well as the hypoxia induced factor 1 alpha (HIF-1 alpha) by Gpx1 [73] results in the antioxidative protective function of TGF-B1, stressing the regulatory role of the ECM.

Therefore, targeting *Gpx1* at an early phase of healing can prevent oxidative stress in hypoxic conditions and in diseases with dysfunctional TGF-B, as osteogenesis imperfecta [74]. Furthermore, the expression of the ribosomal protein (Ubb) is corroborated with the increase in cellular activity in the inflammatory phase. The ubiquitin proteasome complex acts as a governing cause in cell cycle progression [75], while mutations in Ubb are seen at the onset of neurodegenerative diseases [76]. Therefore, the importance of Ubb in tissue regeneration must be further elucidated. However, the colocalization of an UBB-positive signal around blood vessels within the fracture callus is also intriguing (Figure 7b,d,f). However, this might indicate an increased cellular activity in the ECM close to the blood vessel [77].

### Implications and Limitations

The differential gene expression and histological analysis performed in this study showed a cross-talk between the six discussed biological processes. The gene network analysis between selected markers from each process asserted such a cross-talk in bone regeneration and bone homeostasis (Figure 9).

The study implemented microarray analysis, which is a powerful tool to explore differential gene expression between over 96 thousand genes. The analysis of such high-throughput data is complex, and the choice of the cut-off parameters can influence the data interpretation. We applied a stringent cut off of a high fold change ≥|2| combined with a low *p*-value *p* ≤ 0.01. Although this has reduced the number of genes on the final list, it has also excluded important genes. Furthermore, the decision of which markers to perform in histology was influenced by technical feasibility, such as the availability of commercial antibodies for IHC and the availability of specimens. One limitation was the unavailability of D0 and D3 for histological analysis. Another consideration is the use of young mice for the study, as the intact bones are still undergoing bone maturation where some pathways might be more active than others, which might influence the relative expression interpretation.

## 4. Materials and Methods

The current study was conducted in full compliance with institutional laws and German animal protection laws. Animal experiments were approved by the Animal Ethics Committee in the state of Berlin, Germany (“Landesamt für Gesundheit und Soziales, Berlin,” permit number: G0206/08). The animal experiments were performed following the Animal Research: Reporting of In Vivo Experiments (ARRIVE) guidelines [78] to enable the scientific community to reproduce the methods and results.

### 4.1. Experimental Design

C57BL/6N male mice (N = 67) of 8 weeks of age were purchased from Charles River Laboratories (Wilmington, MA, USA) and were randomly assigned to different time points. The animals were housed in standard conditions (2–4 animals/cage). A standard mid-diaphyseal closed fracture was generated in the left femora of animals using the method developed by Bonnarens and Einhorn [79]. The animals were euthanized at (D = days) D0, D3, D7, D10, D14, D21, and D28 postfracture.

### 4.2. Surgical Procedure

The surgical procedures were previously reported. Briefly [21,80], the animals received 2.5% isoflurane/oxygen and a subcutaneous injection of 1 mg/kg Buprenorphine (Reckitt Benckiser, Ludwigschafen am Rhein, Germany) for anesthesia before surgery. Then, the left part of the knee was bent, and an incision was made through the skin to expose the patella. The patella was then moved to the lateral side to expose the femoral condyles. A passage for the intramedullary pin was created through the distal end of the femur using a hollow needle (BD MicrolanceTM, 0.55 mm, sterile, Becton Dickinson Life Sciences, Heidelberg, Germany). This needle was removed, and then an intramedullary pin (Thermo spinal needle 17, 0.5 × 0.9 mm, Thermo Europe N. V., Leuven, Belgium) was inserted into the bone marrow cavity. After relocating the patella, the wound was carefully sutured.

### 4.3. RNA Isolation and Microarray Hybridization

The fractured callus was collected immediately after euthanasia, and then the surrounding soft tissue and the intramedullary pin were removed. The fractured bone samples (N = 5/time point), including 1 mm diaphyseal bone on either side, were taken and immediately snap frozen in liquid nitrogen and stored at −80 °C until further use. 

RNA extraction was performed as previously described [21,80]. Briefly, bones were pulverized using a pestle and mortar and then homogenized in TRIZOL using T10, Ultraturrax, while placed on ice. Total RNA was then isolated using TRIZOL (Invitrogen Life Technologies, Karlsruhe, Germany) as per the manufacturer’s protocol, with DNAse I digestion (Invitrogen Life Technologies, Karlsruhe, Germany) included. A 150 ng of total RNA per sample was pooled in a given group, and then the pooled mixture was distributed as 200 ng samples per tube for technical triplicates.

Whole genome transcriptome profiling was carried out using Illumina’s MouseRef-8 v2.0 Expression Bead Chips (Illumina, Ambion, TX, USA). The processing of RNA for cRNA synthesis and labeling was performed according to the standardized protocol provided by the Illumina TotalPrep RNA amplification kit (Illumina, Ambion, TX, USA). Further steps of hybridization, washing, and scanning of the array were carried out as instructed in the Illumina Whole-Genome Gene Expression Direct Hybridization Assay guide. 

### 4.4. Microarray Data Analysis

The obtained raw data files were analyzed using different packages from R (version 3.0.3) [81] and Bioconductor (version 3.5) [82]. The analysis of IDAT files obtained from Illumina whole genome sequencing was read through the Illuminaio package [83], and background corrections along with data normalization were carried out using the Bead array package [84]. Differential gene expression analysis was carried out using the Limma package [84]. Pearson’s correlation coefficient was calculated to perform a quality assessment of array data. Fold change (FC) was calculated from the mean expression between D0 and other time points. The two-tailed student T-test was used to calculate the significance level. Detailed differential gene expression analysis was performed using FC ≥|2| and *p* ≤ 0.01 cut-offs. 

Hierarchical clustering was performed to align differentially expressed genes (DEG) based on their expression profile across time points. Functional annotation of genes was carried out using the Database for Annotation, Visualization, and Integrated Discovery (DAVID) bioinformatics database version 6.7 [85].

### 4.5. Pathway Analysis

Differentially expressed genes involved in key biological processes were mapped to specific signaling pathways using the Kyoto Encyclopedia of Genes and Genomes (KEGG) database [86]. The specific signaling pathways of interest were TGF-beta signaling, MAPK signaling, and Wnt signaling. 

### 4.6. Network Analysis

The prediction of physical or genetic interactions helps in understanding the complexity of a biological process. Therefore, a web-based interface; GeneMania [87], was used to predict co-expression and colocalization among the DEGs involved in key biological processes. The obtained interactions were then imported into Cytoscape [88] (version 3.4.0) to visualize the network.

### 4.7. Sample Preparation for Histological Examination

Left femora were harvested with surrounding muscle tissue after euthanasia and fixed in 4% paraformaldehyde (PFA) for the histological examinations as described previously [52,89]. The bone samples for histology were collected at D7, D10, D14, D21, and D28. After 48 h of fixation at 4 °C, the decalcification of bone samples was carried out using a 1:2 solution of 4% PFA and 14% Ethylenediaminetetraacetic acid (EDTA) at 4 °C for 3 weeks. The solution was changed every three days. The bone samples were embedded in paraffin post-dehydration and stored at 4 °C for 24 h before sectioning. The blocks were cut into 5 µm thick sections using a motorized rotary microtome (Thermo/Microm HM 355S, Thermo Scientific GmbH, Karlsruhe, Germany).

### 4.8. Histological Stains

The structural and cellular changes during the progression of healing were investigated using the following histological stains: (1) Trichrome Masson Goldner stain [90,91] to obtain an overview of mineralized and nonmineralized bone matrix changes within the fractured callus. (2) Sirius red stain [65] to investigate the changes in type I collagen distribution with the progression of healing. The stain facilitates the visualization of type I and type III collagen under polarized light. However, this study mainly focused on investigating the organization of type 1 collagen using Sirius red stain and silver nitrate stain [52,91] to examine the morphological changes of osteocytes over the course of healing.

### 4.9. Immunohistochemical Stains

Immunohistochemistry (IHC) supported the microarray analysis findings. The decalcified sections were used to conduct the following immunostains: (1) Alpha smooth muscle actin (α-SMA) to investigate the blood vessel formation during healing. Rabbit polyclonal primary antibody in 1:100 dilution was used for α-SMA labeling (TA353561, Acris Antibodies GmbH, Herford, Germany). (2) Matrix extracellular phosphoglycoprotein (MEPE) to investigate bone mineralization and extracellular matrix formation. Rabbit polyclonal primary antibody in 1:100 dilution was used for MEPE labeling (HPA038004, Sigma-Aldrich Chemie GmbH, Munich, Germany). (3) Extracellular signal-regulated kinase (ERK) to investigate the extracellular matrix activity. Rabbit monoclonal primary antibody in 1:100 dilution was used for ERK labeling (ab32081, Abcam Cambridge, MA). (4) Glutathione peroxidase 1 (GPX1) to investigate the mitochondrial activity in fracture healing. Rabbit monoclonal primary antibody in 1:325 dilution was used for GPX1 labeling (ab22604, Abcam Cambridge, MA). (5) Ubiquitin B (UBB) to investigate the ribosomal activity in the fracture healing process. Rabbit polyclonal primary antibody in 1:150 dilution was used for UBB labeling (ab7780, Abcam Cambridge, MA, USA). Briefly, deparaffinized sections were blocked with Bloxall blocking solution (SP-6000, Vector Laboratories, Inc., Burlingame, CA, USA) for 10 min. Afterwards, the samples were incubated with the primary antibody in dilution as stated above in Dako antibody diluent (Agilent Technologies Inc., Santa Clara, CA, USA). The sections were then incubated with secondary antibodies (Alkaline phosphatase, AK 5200, Vectastain ABC kit, Vector Laboratories, Inc., Burlingame, CA, USA) diluted in TBS and universal serum for 30 min at room temperature. The slides were then incubated with vector red solution (Vector Red Alkaline phosphatase substrate, SK-5100, Vector Laboratories, Inc., Burlingame, CA, USA). The α-SMA stained sections were counterstained with methyl green. MEPE, ERK, GPX1, and UBB-stained sections were counterstained with Toluidine Blue. An additional set of GPX1 and UBB-stained sections were counterstained with silver nitrate.

### 4.10. Image Capturing and Histomorphometry

Images were captured using a Leica microscope (Leica DM5500 photomicroscope equipped with a DFC7000 camera, Leica Microsystem Ltd., Wetzlar, Germany) and LASX software version 3.0 (Leica Microsystem Ltd., Wetzlar, Germany). Histological and immunohistochemical-stained slides were imaged at 10× and 40×, and in oil emersion at 100× magnification in bright field. GPX1 and UBB sections that were counterstained with silver nitrate were visualized using a Texas-Red filter (Leica Microsystem Ltd., Wetzlar, Germany) at >560-nm emission wavelength. The excitation wavelength range for the fluorescent red substrate was set in the range of 365–560 nm at 40× magnification. 

Collagen fibrils in Sirius red stained sections were visualized using a Leica microscope, equipped with an additional polarized analyzer, detector (11555079, rotatable; Leica Microsystem Ltd., Wetzlar, Germany), and lambda filter (11513907, Leica Microsystem Ltd., Wetzlar, Germany).

Histomorphometry was performed using a modified trainable weka segmentation plugin [92] from ImageJ software (version 1.52 d, NIH, Bethesda, MD, USA). The count of osteocytes on silver-nitrate-stained slides was measured using the Cell counter plugin of ImageJ.

### 4.11. Computational Segmentation of Collagen Fibers

The monochrome images obtained from the polarized filter were used to conduct the quantitative analysis of collagen fibers. MATLAB Natick, MA, USA (version R2017b)-based standalone package CT-FIRE [48,93] Madison, WI, USA (version 2.0 Beta) was used to computationally segment the collagen fiber properties.

### 4.12. Statistical Analysis

The normal distribution of the data was tested using descriptive statistics using IBM SPSS 24.0 (IBM Corporation, Armonk, NY, USA). A one-way analysis of variance (ANOVA) combined with the Bonferroni correction was used for the parametric distribution. The statistical testing of the nonparametric distribution was performed using the Mann-Whitney U-test combined with the Bonferroni correction. Bar graphs were plotted as a mean ± standard error of mean (SEM).

### 4.13. Functional Enrichment Analysis

Gene ontology analysis for differentially expressed genes associated with six key biological processes was carried out using an R-based package, ClusterProfile (version 4.6.0) [94]. The top 10 enriched biological processes were obtained for each of the gene sets and visualized as bar plots.

## 5. Conclusions

The presented results demonstrate the complexity, coordination and overlap in gene regulation between different biological processes that achieve successful bone healing. The study showed for the first time the importance of oxidative stress regulation through an interplay between mitochondrial and ribosomal genes as well as the extracellular matrix proteins. This role shall be addressed further to better understand the underlying mechanism, especially in relation to the role of the immune response.

## Figures and Tables

**Figure 1 ijms-24-03569-f001:**
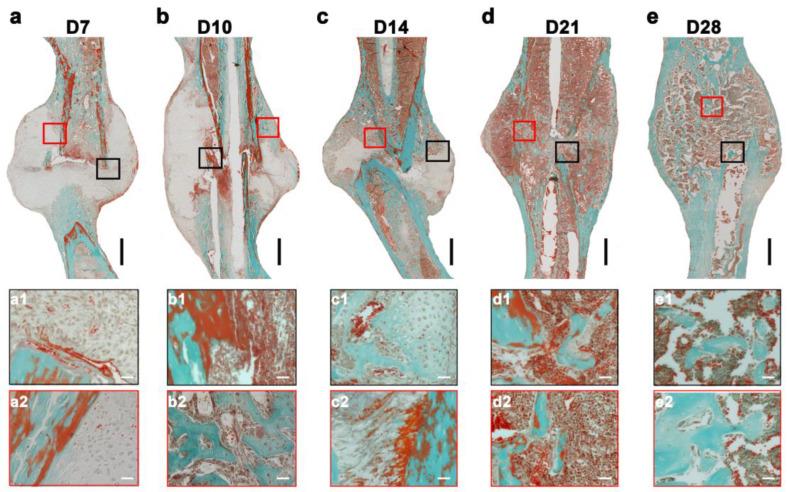
Assessment of bone matrix mineralization at the fracture site reflects the healing pattern. Trichrome Masson Goldner staining of the fractured callus differentiated mineralized (green) and nonmineralized (red) portions of intramembranous bone (red rectangle) and endogenous bone formation (black rectangle). (**a**, (**a1**,**a2**)) A higher portion of the nonmineralized matrix was seen in the fractured callus at D7. (**b**, (**b1**,**b2**)) The patches of the mineralized matrix were seen in the periosteal regions within the callus at D10. (**c**, (**c1**,**c2**)) A lower number of chondrocytes and a larger area of the mineralized matrix were seen at D14 compared with earlier time points. (**d**, (**d1**,**d2**)) The advancement in bone matrix mineralization was evident by D21. (**e**, (**e1**,**e2**)) Bony bridging and complete mineralization of newly formed bone were seen by D28. (Green: mineralized bone, red: nonmineralized bone; scale bar: upper panel (250 µm), lower panel (25 µm)).

**Figure 2 ijms-24-03569-f002:**
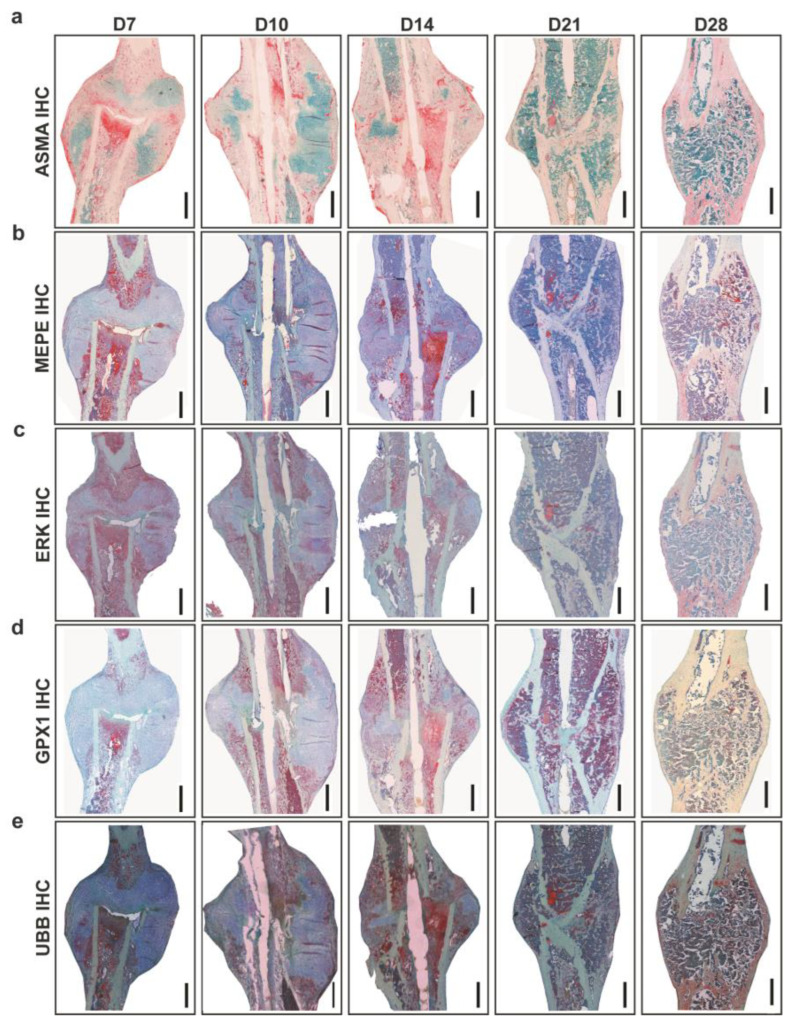
Immunohistochemical stains depict the changes in the fractured callus across the time points of healing. (**a**) ASMA IHC stain helped in the visualization of blood vascularization. (**b**) MEPE IHC, and (**c**) ERK IHC stain helped in the cellular understanding of ECM regulation. (**d**) GPX1 IHC stain was carried out to examine mitochondrial activity at the cellular level. (**e**) UBB IHC stain was carried out to examine ribosomal activity at the cellular level (scale bar: 250 µm).

**Figure 3 ijms-24-03569-f003:**
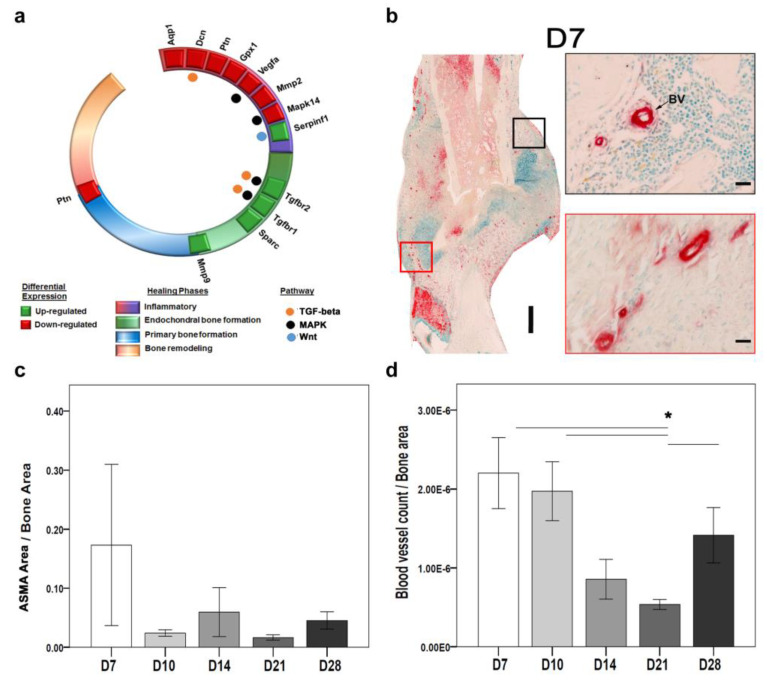
Early inflammatory phase correlates with angiogenesis at both the molecular and cellular level in wild-type mouse. (**a**) Differential gene expression analysis showed that a higher number of angiogenesis genes were downregulated during the inflammatory phase. (**b**) ASMA-positive–stained regions were seen around the fractured gap most abundantly at D7. (**c**) Quantitative analysis showed the highest ASMA-positive area at D7 and the lowest at D21. (**d**) Significantly lower blood vessel counts at D21 compared with all other time points. (N: D7 = 4, D10 = 6, D14 = 3, D21, D28 = 5; nonparametric distribution, Mann-Whitney U-test, * = *p* ≤ 0.05, red and black boxes represent higher magnification, BV: blood vessels, scale bar: (**b left** panel): 250 µm, (**b right** panel): 25 µm)).

**Figure 4 ijms-24-03569-f004:**
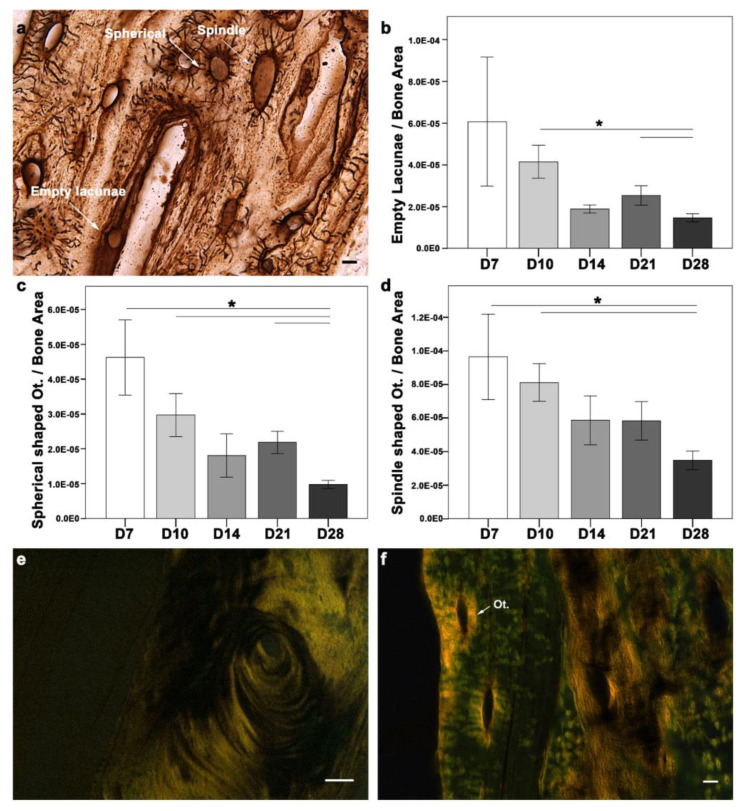
Histological depiction of morphological changes in osteocytes and collagen fibril arrangement during fracture healing in the fracture gap. (**a**) Osteocytes were present in abundance around the fractured callus at all time points (spindle: active, spherical: intermediate, empty: dead). (**b**) Empty lacunae were significantly lower at D28 when compared with D10 and D21. (**c**) Spherical-shaped osteocytes were significantly lower at D28 when compared with D7, D10, and D21. (**d**) Spindle-shaped osteocytes were significantly lower at D28 when compared with D7 and D10. (**e**) Well-arranged collagen fibers seen around the fractured callus (bright yellowish areas) (**f**) Type I collagen ring-like structure seen around the osteocytes (bright yellowish areas); positive stained regions were located within the ossified region at the periosteal surface of the callus tissue at D7. (N: D7 = 4, D10 = 6, D14 = 3, D21, D28 = 5); non-parametric distribution, Mann-Whitney U-test, * = *p* ≤ 0.05, Ot: osteocyte, scale bar: (**a**) (5 µm), (**e**) (25 µm), (**f**) (5 µm)).

**Figure 5 ijms-24-03569-f005:**
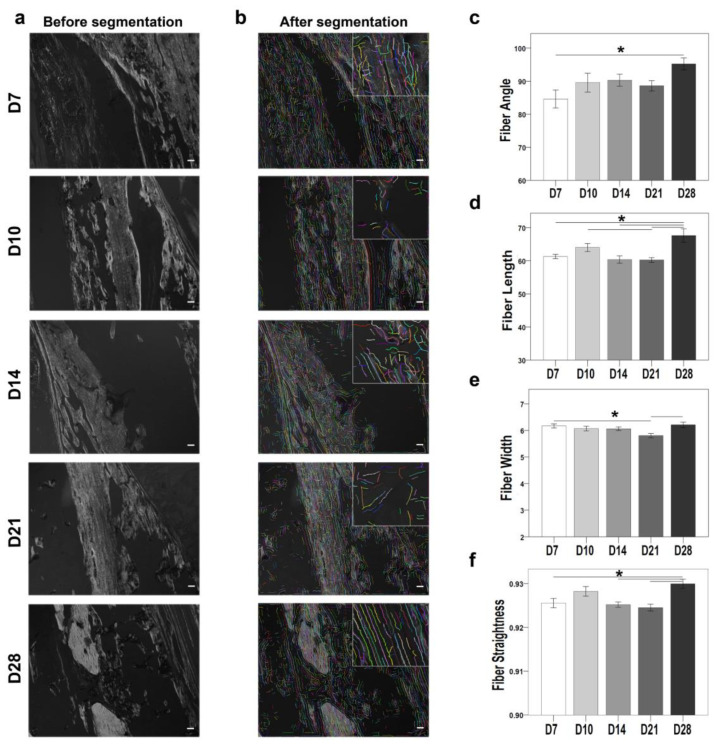
Quantitative evaluation of collagen fibril properties using the CT-FIRE plug-in showed enhanced bone quality at D28. CT-FIRE, a standalone MATLAB-based tool, was used to quantify the collagen fibers from polarized microscope Sirius red-stained images. (**a**) Before segmentation; (**b**) After segmentation. The right upper side shows the enlarged collagen fibers (**c**–**f**) D28 showing higher fiber angle, length, width, and straightness compared with D7, D10, D14, and D21. (N; D7 = 4, D10 = 6, D14 = 3, D21, D28 = 5; nonparametric distribution, Mann-Whitney U-test, * = *p* ≤ 0.05, scale bar: 50 µm).

**Figure 6 ijms-24-03569-f006:**
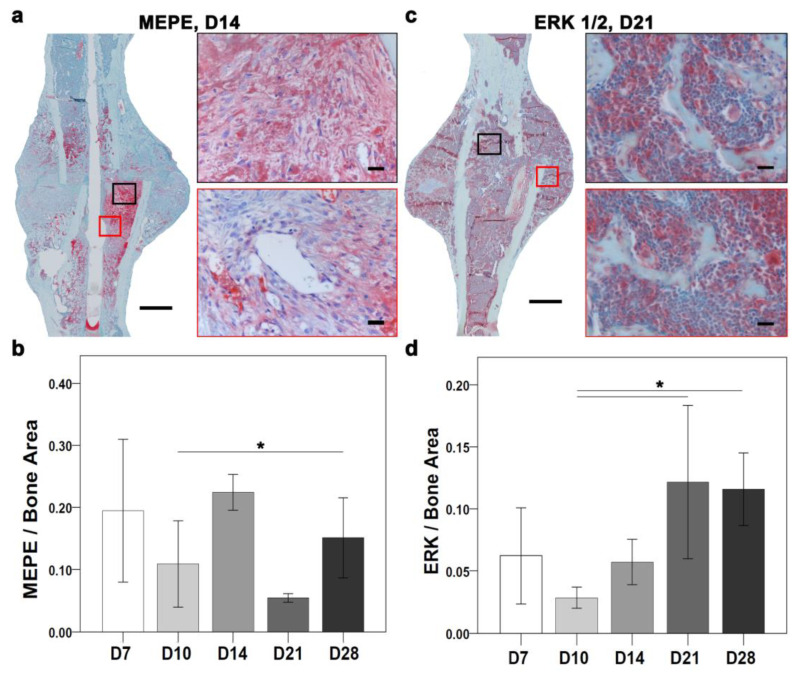
MEPE and ERK activity are directed toward the establishment of bone homeostasis during bone healing. (**a**) MEPE-positive signal was most clear around the callus and within the mineralized matrix at D14. (**b**) Quantitatively, MEPE-positive area was highest at D14 and lowest at D21. D10 showed a significantly lower MEPE-positive area compared with D28. (**c**) ERK-stained regions were distributed around the whole callus at D21. (**d**) ERK-stained area had the highest portion at D21 and the lowest at D10. However, the ERK-positive area was significantly lower at D10 compared with D21 and D28. (N: D7 = 4, D10 = 6, D14 = 3, D21, D28 = 5; red and black boxes represent higher magnification, non-parametric distribution with Mann-Whitney U-test; * ≤ *p* = 0.05, scale bar: (**a**,**c**) (**left**: 250 µm, **right**: 25 µm)).

**Figure 7 ijms-24-03569-f007:**
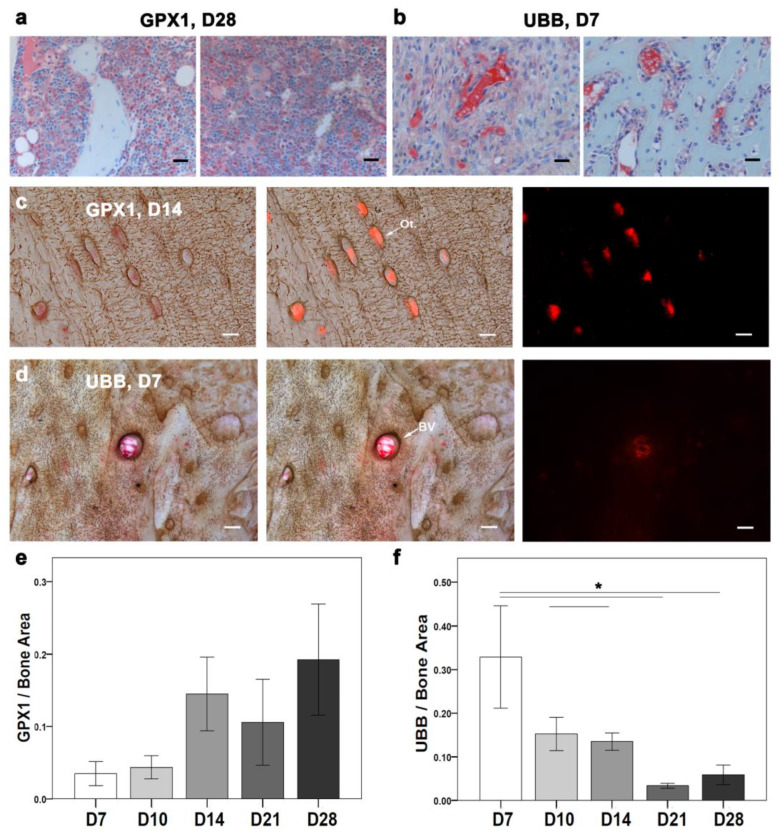
Temporally contrasting regulation of mitochondrial and ribosomal genes reflected on signal intensity and localization of GPX1 and UBB proteins through healing progression. (**a**) Mitochondrial marker GPX1 signal was abundant at D28 and localized around the cells in the newly formed bone and within the bone marrow. (**b**) Ribosomal Marker UBB signal was aberrant at D7 and localized in cells at the periosteal region and within bone marrow cells. (**c**) GPX1-positive signal was detected within active osteocytes. (**d**) UBB-positive signal was visualized around blood vessels in the fractured callus; however, none was seen within the osteocyte or their vicinity. (**e**) GPX1-positive signal portion was highest at D28. (**f**) UBB-positive signal portion was significantly higher at D7, D10, and D14n compared with D21. (Ot: osteocytes; N: D7 = 4, D10 = 6, D14 = 3, D21, D28 = 5; nonparametric distribution with Mann-Whitney U-test; * ≤ *p* = 0.05; scale bar: (**a**,**b**) (25 µm), (**c**,**d**) (5 µm).

**Figure 8 ijms-24-03569-f008:**
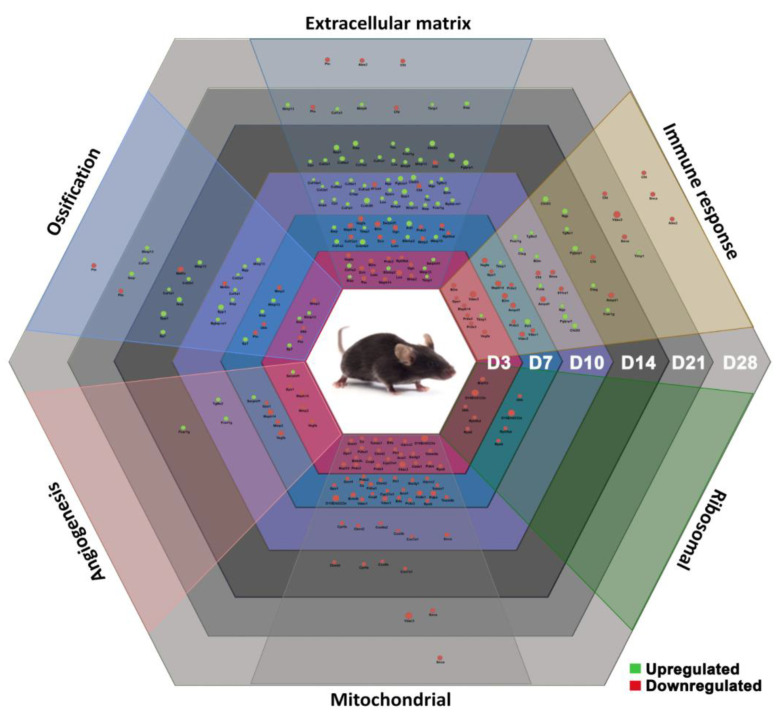
Hexagon summarizes the temporal distribution of differentially expressed genes involved in immune response, angiogenesis, ossification, ECM regulation, mitochondrial, and ribosomal activity.

**Figure 9 ijms-24-03569-f009:**
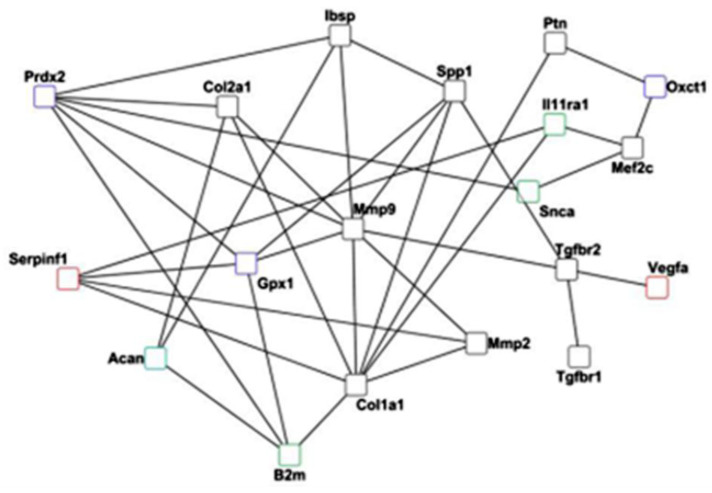
Network analysis carried out using GeneMania depicts the cross-talk between differentially expressed genes.

**Table 1 ijms-24-03569-t001:** List of differentially expressed pro- and anti-inflammatory genes and their FC and *p*-value (white background: downregulated and grey background: upregulated).

	Gene	D3	D7	D10	D14	D21	D28
Proinflammatory	*S100a8*	−4.5539	−4.54909				
		5.83 × 10^−6^	5.16 × 10^−6^				
	*B2m*	−3.156	−3.394	2.04			
		1.63 × 10^−4^	4.56 × 10^−7^	2.49 × 10^−5^			
	*Rsad2*					−2.7446	−2.333
						1.45 × 10^−6^	7.18 × 10^−6^
	*Il11ra1*			−2.3639			
				3.09 × 10^−6^			
	*Snca*			−2.182		−3.173	−3.102
				7.69 × 10^−6^		2.38 × 10^−6^	1.46 × 10^−5^
	*Ctsg*			2.769	2.342		
				2.01 × 10^−5^	2.32 × 10^−5^		
	*Lcn2*			4.061	3.698		
				1.56 × 10^−6^	2.76 × 10^−6^		
	*Pgyrp1*			2.73	2.361		
				3.78 × 10^−4^	7.2 × 10^−4^		
	*Fcnb2*			2.521			
				1.13 × 10^−5^			
	*Fcer1g*			2.316	2.598		
				9.37 × 10^−6^	1.81 × 10^−6^		
Anti-inflammatory	*Ampd1*			−2.3216	−2.5181		
				1.88 × 10^−5^	3.83 × 10^−6^		
	*Ifitm2*			2.008			
				3.87 × 10^−5^			
	*Slpi*			2.3009			
				1.16 × 10^−5^			

## Data Availability

Data will not be available due to privacy restrictions.

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
