# Peer review of "Landscape of Well-Coordinated Fracture Healing in a Mouse Model Using Molecular and Cellular Analysis"

_ijms, 2023, doi:10.3390/ijms24043569_

Round 1

Reviewer 1 Report

The authors tried to detect the complex interactions in gene regulation among different biological processes during fracture healing via whole genome transcriptome and histological analysis. I think it is an interesting work. There are some comments as follows:

Major comments:

1.       One of the main objectives of this study was to analyze changes in different genes at different points post fracture. However, starting from Figure 2, the authors only listed the pictures of major time points of change, which made it difficult to give the reader a sense of dynamic change. It is suggested to list pictures of histological staining during all the time points. Moreover, in Figure 2, the authors showed the changed genes of angiogenesis. It is also suggested to list the dynamic changes of these genes across all the time points. The following results should be presented in the same way.

2.       Since the author wanted to observe the dynamic synthesis of genes, it is suggested that they can discuss all the changes of genes together according to the time point of bone healing, so that readers can more intuitively understand the changes of each gene in the whole process of repair.

3.       Much of the description in the results section cannot be found in the chart, especially the changes in the genes involved.

Minor comments:

1.       Why were there different numbers of samples at each time point.

2.       The picture was not clear enough and looks blurry.

3.       There were many type errors. For example, there were two periods at the end of sentence in line 230.  In line 529-530, there were type errors of Negfa.

Round 2

Reviewer 1 Report

There were no more comments.

Author Response

Dear Reviewer,

Many Thanks for your prompt reply, and constructive comments.

Reviewer 2 Report

The authors revised the manuscript and addressed well the issues in the previous manuscript. There are two minor errors are required to be corrected.

1. On page 4 line 1103, there are two ". ." in front of later word.

2. On page 26 line 821, "After 448 hof fixation" should be corrected.

Author Response

Dear Reviewer,

Many Thanks for the detailed review, constructive comments, and prompt review.

we have corrected the minor errors indicated by you.

Kind regards

The authors